# Big Data in Biodiversity Science: A Framework for Engagement

**Tendai Musvuugwa [1], Muxe Gladmond Dlomu [1] and Adekunle Adebowale [2],***

1    Department of Biological and Agricultural Sciences, Sol Plaatje University, Private Bag X5008,
     Kimberley 8300, South Africa; tendai.musvuugwa@spu.ac.za (T.M.); dlomumuxe@gmail.com (M.G.D.)
2    Department of Botany, Rhodes University, Grahamstown 6140, South Africa
*    Correspondence: strychnos009@gmail.com

**Abstract:** Despite best efforts, the loss of biodiversity has continued at a pace that constitutes a major threat to the efficient functioning of ecosystems. Curbing the loss of biodiversity and assessing its local and global trends requires a vast amount of datasets from a variety of sources. Although the means for generating, aggregating and analyzing big datasets to inform policies are now within the reach of the scientific community, the data-driven nature of a complex multidisciplinary field such as biodiversity science necessitates an overarching framework for engagement. In this review, we propose such a schematic based on the life cycle of data to interrogate the science. The framework considers data generation and collection, storage and curation, access and analysis and, finally, communication as distinct yet interdependent themes for engaging biodiversity science for the purpose of making evidenced-based decisions. We summarize historical developments in each theme, including the challenges and prospects, and offer some recommendations based on best practices.

**Keywords:** big data; biodiversity; data curation; data generation; cyber infrastructure; data access; science communication

## 1. Introduction

Biodiversity refers to the variety of genes, species and ecosystems of life on Earth, and is the source of many essential goods and services (e.g., food, timber, medicine, nutrient recycling, crop pollination) that support human well-being and quality of life [1]. Despite several international treaties, efforts and commitments to curb its loss, biodiversity continues to decline at a rate above species discovery rate, largely due to anthropogenic factors [2]. To assess the status and trends (local and global) in biodiversity requires a vast amount of relevant information on the distribution and abundance of different species across varying spatial and temporal scales [3]. In other words, relevant data need to be collected, collated, and analyzed.

The last two and half decades have witnessed an exponential increase in the generation and analysis of data in virtually all domains of human engagement such that the term 'big data' was coined to distinguish the data explosion era from what went on before [4,5]. Scholz (2017) [6] tracked the origin of the term to the 1960s and 1970s and summarized its appearances in documents from the US Congress publications to various academic and non-academic works spanning a period from 1961 through to 1979. These early usages had little bearing on how it is conceived today. In its more contemporary form, several authors, for example, [5,7], have traced the emergence of the term from the world of commerce, whose main interest in big data was, and still is, driven by the need to monitor and improve performance. The concept has since spread to several areas of endeavor including, but not limited to, the healthcare industry, the agricultural industry, the education industry, the media sector, governance, the banking and finance sector, astronomy, climate change and biodiversity management. As with concepts of such diverse application, and to which several distinct domains can lay claim, there is no universally satisfactory definition of big data. However, there is a consensus as to the key elements of its essence: big data

is characterized by the three Vs of huge volume, high velocity and diverse variety [5,8]. The volume component refers to the size of data generated, considered in petabytes or higher units of data; the velocity component suggests a rate of generation that is real-time or nearly so, thereby contributing to the huge volume; and the variety indicates a mixture of structured, semi-structured and unstructured pieces of information [4,9]. Two other possible Vs, veracity and variability, are sometimes included and are addressed in various forms later in this paper.

Within the context of biodiversity, big data is defined as a "techno-political tool to manage the distribution of biological species", and as "the intensive data accumulation of digitized information on biodiversity, corresponding to a spatial and temporal description of species distribution" [10]. While these rather similar definitions are limited in their scope, because they ignore some other aspects embedded in biodiversity [1], the first part, nevertheless, provides a historical anchor for situating the deliberate integration of big data and biodiversity within a techno-political agenda. This agenda, which could be viewed in simple terms as the implementation of policy supported by an evidence base, started in the mid-1960s [11], much in the tradition of the data-intensive research of the physical sciences (e.g., The Manhattan Project). Big data is central to biodiversity science because, at its barebone level, biodiversity involves species and their distributions across space and time. For instance, 36.5% of global plant species are considered as "exceedingly rare" [12], suggesting a need for conservation planning to, at least, take such metrics into account. In this paper, we assume the view that Biodiversity Big Data (BBD), as a concept, encompasses a cyclical scheme that involves the generation, curation, processing, analysis and communication of biodiversity information, at huge volumes and diverse varieties, with the purpose of making an informed decision for biodiversity management.

The emergence of big data (BD) as a discipline has raised some philosophical questions, challenging established ways of knowing in the various domains of knowledge, including biodiversity science. Some advocates of BD [13,14] have been quick to declare the end of theory; oppose the need for model building or hypothesis formulation as the sheer size of available data and the power of data analytics allow for pattern detection and the emergence of new insights independent of human bias. This view has been robustly contested in the BD literature, and is shown to be based on fallacious thinking, whilst recognizing the inherent potential of analyzing vast amount of data. Kitchin (2013) [4] and others [5,12,15,16] have shown that BD, however exhaustive, is still representational (a sample) and is therefore subject to the vagaries of sampling bias. Data collection and analysis are shaped by the theories underpinning the systems of collection and the algorithms of analytics. The emergent patterns are, thus, not free of human bias as they are interpreted within frameworks. In addition, there is the real possibility of random correlations between variables with no underlying causal linkage. Succi and Coveney (2019) [17] suggest that the pattern recognition power of BD analysis could provide a basis for further engaging theories in making sense of the patterns that would be otherwise undiscernible to the human mind. One application of BD raised in the work [17], and which is relevant to biodiversity science, is its ability to handle some of the sensitive aspects of non-linearity or chaos found in many complex systems [18]; a concept that underlies spatio-temporal organization and weather events and is best encapsulated by the popular phrase 'the butterfly effect'.

In addressing "the datafication of biodiversity" [10], it was convincingly demonstrated that the process of transforming ecological and other records of living forms into biodiversity data not only changes the nature of the information, it also corresponds to a politically-driven shift in priorities for ecological research from local concerns to a global outlook, resulting in the birth of global biodiversity. The key element was to underpin sustainability policy with a strong evidence base. They highlighted the positive role played by the creation of the global biodiversity information facility (GBIF)—one of the largest biodiversity databases in the world—in bridging the divide between science and politics for the global good. The aim was to facilitate the translation of good science to good

government policy [19]. While this datafication process provided one approach to viewing the global environmental landscape and developing some of the tools for effective monitoring, it nevertheless came at the expense of biological context. As argued by Bowker (2000) [20], BBD production often results in the loss of ecological meaning as species become disconnected from their ecological context in the process of achieving uniformity and compatibility of data format in a single database. This poses a peculiar danger of database creation becoming an end in itself [20]. A similar line of reasoning was extended further by detailing how the real-world ecological niche of various organisms, captured by numerous information records, are reduced to a two-dimensional world of "rows and columns" [10], thereby creating an artificial data niche detached from the biophysical realities of the organisms supposedly represented. This is what was construed as the datafication of ecological records [10].

The discourse around the emergence of BD in biodiversity would be incomplete without consideration for the infrastructures that make it possible to generate, store and analyze BD. These infrastructures vary from instruments capable of recording tens of petabytes of information (e.g., radio-telescopes) to next-generation sequencers for sequencing whole genomes (about 3 billion nucleotide base-pairs in one human being for instance), to remote sensing devices for collecting a vast amount of environmental data. In addition to these are the rapidly increasing computer storage capabilities, including storing in the cloud, increasing computational power of PCs, coupled with innovations in statistical computing that allow new ways to analyze and visualize BD.

In this review, we adopt the life cycle of data as a framework to interrogate four main objectives, which are to (1): summarize the current state of BBD under each theme of the scheme, (2) identify opportunities for innovation/collaboration, (3) identify challenges, and (4) propose recommendations to drive best practices in the business of BBD. Figure 1 presents a schematic of the life cycle of biodiversity data as applied in this paper. Starting from data acquisition/generation, we track the journey of BBD and associated events through storage/curation, data access, data processing and analysis, and finally communication and decision making. An important point to note in all this is the reusable nature of BBD to address diverse questions relevant to the field. In treating each theme, we attempt to track practices from the past, through to the present, and where possible anticipate the direction for the foreseeable future.

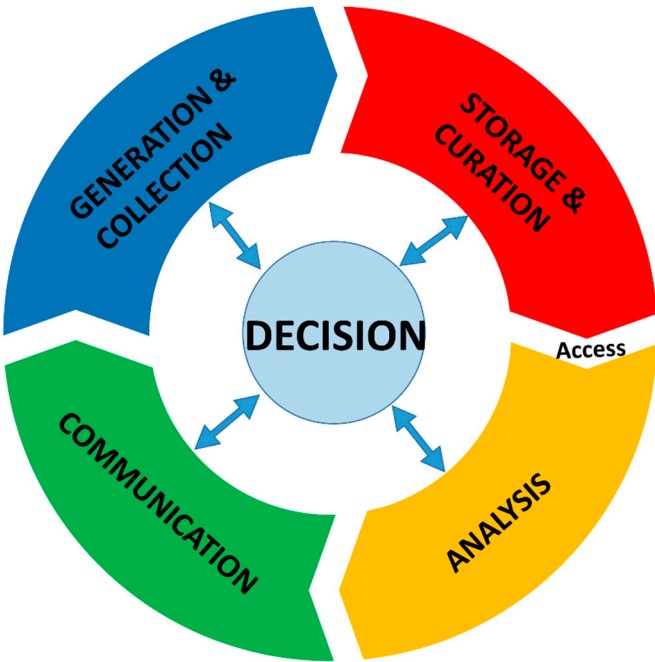

**Figure 1.** Schematic framework of data life cycle for engaging biodiversity science.

## 2. Data Generation and Collection

Biodiversity data generation and collection encompasses the various procedures, technologies and methodologies deployed to create and collate biodiversity-relevant datasets for subsequent use in the data value chain. The collation, integration and analysis of massive datasets has grown rapidly with advances in enabling technologies and infrastructures on the one hand, and the need for regional and global scale ecological assessment and monitoring on the other. It could be argued that the need for big biodiversity data collection and the technologies to achieve such ends have mutually enhanced each other in their sophistication. The importance of BBD collection is underscored by widely acknowledged pressures on biodiversity through loss and habitat degradation, the need to plug data gaps and to develop efficient monitoring initiatives aimed at informing scientists, conservation managers and the general public on the state of global biodiversity [21,22]. Furthermore, the possibility of coupling biodiversity metrics with the sustainability agenda adds robustness to ecological forecasting, whilst providing near real-time status of the environment [23,24].

### 2.1. Types of Biodiversity Data and History of Their Collection and Collation

Measuring biodiversity is a difficult undertaking due to the complex nature of biodiversity itself [25]. Depending on the questions of interest and the spatial and temporal scale of reference, biodiversity data types can vary from the taxonomic, the biogeographic to functional traits [26], and include molecular data (e.g., DNA sequences), species occurrence data, remotely sensed data of various forms (e.g., vegetation cover from satellite imagery) [27]. Traditionally, biodiversity data were collected without much standardization and were usually based on limited observations with little thought to notions of repeatability and statistical powers [25]. Often, data were captured in spreadsheets, small disaggregated local databases with little or no interoperability [27] or presented as volumes of floras and checklists sitting on library shelves. However, the last few decades have witnessed a progressive shift from the traditional methods. Data may now be generated and collected through automated instrumentations; static datasets sitting in bookshelves and as physical specimens in museums are now being digitized and made available online; databases and related infrastructures are being developed with an interoperability outlook for data aggregation on a grand scale [28,29].

Biodiversity monitoring, which requires repeated measurement of the same set of parameters over time, and the capacity to automate such process has meant that previous data collection and integration practices would inevitably be disrupted by novel approaches. The new approaches, which includes DNA sequence data generation, remotely sensed environmental data, aggregation of ecological data, integration, and mobilization of species occurrence databases across geographical boundaries, have all evolved hand-in-hand with developments in computer processing powers. A class of biodiversity data made readily accessible by the new approaches are metadata. Their value in biodiversity conservation is only just being recognized and appreciated. These are especially relevant in the taxonomic domain and may include such details as the names of specimen collectors, other locations where voucher materials are stored and date of collection, among others. With regards to museum and herbaria collections, metadata are being used to map a time series of phenological and other important biological events to track biodiversity responses to climate change [30]. They are also providing insight into how biodiversity datasets are being utilized.

### 2.2. Sources and Repositories of Big Biodiversity Data

Taking the species as a convenient unit of biodiversity, information regarding its taxonomy, distribution range, genetic diversity, population structure, community and ecosystem functions, and its adaptability to the abiotic components of the landscape all become relevant sources of biodiversity data for the species. Collecting such information on one individual carries little significance. However, when such data are scaled to thousands

or even millions of individuals, across taxonomic categories and spatial and temporal scales, the outcomes are inevitably big data on which important biodiversity decisions could be based [31]. Although continuously acquiring new data is ideal, this is not always possible due to time and monetary constraints, thus leading to the use of existing datasets housed as collections in museums and herbaria, or published literature and other online platforms of private individuals, governments and NGOs [32].

A number of studies have highlighted the potential of biodiversity collections and their associated metadata held in museums and herbaria to improve biodiversity management plans for example, Reference [30]. The digitization of these collections is still in its early phase. Some reports suggest an estimate of less than 10% of collections in museums and herbaria are available in digital form [33]. Given that there are well over a billion specimens in thousands of collections around the world [34], even a modest number of 5% digitized records constitutes an impressive source of data for biodiversity assessment for generating robust predictive models.

Data repositories can serve dual purposes in biodiversity research: they act as storage platforms (both physical and digital) for data until required for use; they also serve as sources of data to be mobilized and integrated with other compatible platforms for a more in-depth analysis. While there are many online repositories of biodiversity data (see Appendix A Table A1), two, in particular, provide good illustrations of the volume of data available. The GenBank is a genetic sequence database of the National Institutes of Health based in the United States of America [35] and is part of the International Nucleotide Sequence Database Collaboration. Its current release version 242.0 (https://www.ncbi.nlm.nih.gov/genbank/statistics/, accessed on 10 May 2021) has 12.27 trillion nucleotide bases if one includes whole genome data, and well over 2 billion sequences. The numbers keep growing daily. The Global Biodiversity Information Facility (GBIF) is an international network and research infrastructure with a focus on providing free and open access biodiversity data to everyone. The platform currently holds nearly 1.7 billion occurrence records from about 60,000 datasets from across the globe (https://www.gbif.org/, accessed on 10 May 2021). The taxonomic breadth, genetic depth and geographic/ecological scale of data coverage represented by these two, and other similar repositories, has placed enormous amount of information within the reach of biodiversity researchers.

A last, but not the least, source of data collection is through citizen science. Citizen science initiatives involve enlisting members of the public to gather information, which is then pooled for analysis [36,37]. Advances in the development of smart mobile technology has further made data collection through this approach more amenable to biodiversity research [38]. Several citizen science drives around the globe, sometimes in the form of bioblitz, continue to contribute data to the large scale monitoring of biodiversity. The monitoring of charismatic taxa such as birds has benefited, in particular, from the efforts of citizen scientists spread all over the world. Finally, the development of online platforms (e.g., iNaturalist; Flickr and some social media) where amateurs and experts alike can readily interact and contribute data such as geotagged images of organisms, for screening and identification as needed, is a further data resource relevant to biodiversity. For a list of other potential data sources applicable to biodiversity, including their perceived strength and weaknesses, see an excellent summary in [31].

### 2.3. Data Collection Planning

From the Data Information Knowledge Wisdom (DIKW) hierarchy [39,40], it is clear that data are the foundation for information, knowledge and ultimately wise decision-making. The planning processes leading to data generation and collection are therefore important for quality assurance of the data and for achieving the goal of the undertaking. Although opportunistic data collection has its place and may not be discounted in biodiversity research [41], it is well recognized that deliberate planning, resulting in the systematic generation and collection of data, is the more sustainable approach. This is particularly

so because data collection is expensive, and good quality datasets may be reused either independently or by integration with other types of data. Proper planning prior to data collection can help to identify current data gaps and needs relevant for the achievement of specified biodiversity outcomes. This way, scarce resources can be deployed efficiently, further reducing the cost of data collection.

Because good biodiversity datasets are akin to a gift that keeps on giving, quality assurance measures to ensure accuracy and reliability are necessary components of the data collection plan. In summary, a good data generation/collection plan would include preliminary analysis to identify data needs; identifying goals the data would be contributing towards achieving; defining the steps needed to achieve these goals; and finally selecting the tools and methods best suited to acquire the information or extract the data [31].

### 2.4. Opportunities in Data Collection

Data generation results in a large amount of information on biodiversity that can be accessed for use by almost anyone where access is free. This is desirable in situations where data collection is not feasible, for example, due to financial and time constraints or where one needs to use data collected over a period of many years. The rate at which data is generated has greatly increased with improvements in technology. In some cases, huge amounts of data can be produced within a short space of time as it is no longer done manually. A very good example is the Square Kilometer Array (SKA) project being co-hosted by Australia and South Africa, in which large amounts of astronomical data with a wide array of applications can be amassed within a short space of time.

### 2.5. Challenges in Data Collection

As shown above, data are the basis of science. However, the new scale of collection and synthesis requires a new way of collecting and storing data—with a longer-term view in mind. In some cases, there is little to no training given to data collectors, which results in errors during data collection. This can be the case when data are collected through citizen science efforts where members may not be well trained, for example when one is not able to distinguish between species. This can lead to inaccuracies during data entry resulting in data not being fit-for-purpose. Quality assurance is thus a major challenge to data collection playing out through the skill level of the data collector and the reviewers.

There are situations where there is lack of proper guidelines and procedures to be adhered to when data is collected for certain groups of species or for a given data platform. This leaves each data collector to use their own methods to gather data, which creates huge inconsistences. Some collectors gather data in a biased manner, prioritizing certain areas over others, opportunistically choosing places where they expect to find what they are looking for, or areas easier to navigate. This creates problems for example, when one needs to use that data for comparison purposes, or when the data has been collected using different methods that lack consistency.

Biodiversity data generation can be very time consuming, expensive and labor intensive for one to gather enough data that can be analyzed and confidently used for decision making purposes. Data collection requires good funding, which is often highly competitive and inadequate for the need. This is a bigger challenge in developing countries, which are usually the habitats to many biodiversity hotspots. Thus, the areas in need of biodiversity data are also the most impoverished leading to an asymmetry in the potential and actual amount of data generated from such areas. The data gap has a ripple effect on the quality of research-driven conservation and allied decisions in such regions of the world.

### 2.6. Recommendations in Data Collection

There is a need to ensure that adequate and relevant training is given to data collectors to improve the accuracy and reliability of the collected data. Although it is being done, the quality assurance of data could be improved by developing protocols to guide the data collection process in order to promote consistency and accuracy. Newly generated data



could be routinely evaluated for accuracy based on well-set criteria before approval and storage. Currently, there are efforts in place to channel funding towards generation of biodiversity data in developing countries. However, more still needs to be done in this regard to reduce the asymmetry in potential and actual data generated and collected.

### 3. Data Storage and Curation

There are several types of platforms, some physical and some digital, on which big data on biodiversity is stored and curated. Some of these storage platforms include DNA barcoding databases, image libraries, natural history museums and herbaria, species interaction databases, government departments databases, non-governmental organization databases as well as trait information databases among many others [28,42,43]. Specific examples of the above data platforms include The Open Tree of Life (blog.opentreeoflife.org, GenBank), Barcode of Life Data System (BOLD), Global Biodiversity Information Facility (GBIF; gbif.org), Integrated Digitized Biocollections (iDigBio) [33,44], The Atlas of Living Australia, and various Natural History collections from museums around the world (some of these are listed in Appendix A Table A1). Knowledge of such storage databases is essential for those who utilize the stored data as well as those who generate the data. It is useful to know what these databases are, what kind of biodiversity information they store and the magnitude of data they hold. It will also be of interest to take a closer look at how these databases and methods of storage have evolved over time. Stored big data on biodiversity have been widely used for research purposes, resulting in well-informed decisions on the conservation of biodiversity. To sustainably support excellent research, it is ideal that such data be stored in a well-organized manner [28]. Storage of big data should be done in a credible manner, which ensures that a universally accepted standard is maintained in curating and archiving the data. A great example to achieve this will be the use of a taxonomic framework. There are even increased calls from the scientific community for data aggregators and servers to use tools that enable improved data quality storage at source level [45].

For the most part, biological data have been stored as large and complex datasets that have proven to be very challenging when it comes to the use of such datasets. Some of these storage platforms have, however, evolved over time and went through many developments and improvements to keep up with changing technology and needs of the user communities. Natural History Museums and Herbaria from different parts of the world constitute a reservoir of big data on biodiversity, some of which has been collected, curated and stored for hundreds of years. This has been one of the major biodiversity data storage methods used historically but is also still useful today for many researchers. Examples of such repositories include the Royal Botanic Garden Kew, The Netherlands National Herbarium and several other Natural History Museums located around the world. Invaluable data in these museums and herbaria were largely inaccessible to many interested parties in the past because researchers had to physically visit the respective repositories. However, with the advances in digitizing museum collections, a lot of rich data are being generated from historical specimens, and these are now within the reach of anyone with internet access [29]. The advantage of this system is that digitalized data not only have information on the distribution of species, but also form a connection with other relevant biological data for the species such as phylogenetic and DNA information. For example, in the United States, Integrated Digitized Biocollections (iDigBio) [33,44], which is one of the databases housing big data on biodiversity, serves as the national center for the digitization of biodiversity collections and other related data [29]. Having such data online makes them more accessible compared to being confined to physical specimens only available in museums. Given the pace at which museums are moving their collections online for better accessibility and the values being derived from such efforts, it is expected that virtual museums are here to stay as they complementarily extend the reaches of the physical collections to audiences beyond the physical confinement of the museums themselves. A recent estimate from 3400 global herbaria (Index Herbariorum),

indicates there are about 350,000 plant species, and well over 350 million specimens [46,47]. This is a rich biodiversity minefield of stored big data, whose full potential can be explored if the data is fully digitized and if suitable software platforms are developed to integrate and analyze them.

Clearly, digitization is playing a big role in how data is stored, not only in the conversion of museum and herbaria records as explained above, but also on most other data storage platforms. With the current wave of the fourth industrial revolution, digitally migrating data provides a means of rendering them more accessible to the community. Inevitably, there has been an increase in the volume of data digitally stored and curated. Parallel to the increase in digitally stored data are initiatives that encourage data storage platforms, which store similar kind of data to build collaborations to improve the data quantity and quality through pooling databases rather than having small stand-alone databases that address the same problems [28]. Building standard protocols on data storage and curation that ensure high accuracy, consistency and reliability seems then to be the focus going into the future. Combining data from different sources into one database may require ontological adjustments to ensure harmonization of the information from the different collections into a common platform.

The volume of data on the different platforms has significantly increased over time. The rate at which data are collected and stored has increased considerably in the last decade, in particular. For example, there has been a quantum increase in the amount of molecular data stored in nucleotide sequence repositories such as GenBank and Barcode of Life Data System (BOLD). These databases store billions of DNA sequence data and relevant metadata, including specimen images (for BOLD) on different species. The recent initiative of targeting for sequencing well-curated and identified specimens in natural history museum collections has proven to be one of the ways to quickly generate large amounts of genetic data for storage on these platforms [48]. This is advantageous because it then creates a link between the DNA sequences stored on GenBank for example, to the specimen stored in some natural history museum [48]. As a whole, the advances and changes that have occurred in DNA barcoding, metabarcoding and genomic technology over time have led to the rapid growth of the databases holding such data. These techniques are expected to continue evolving with further improvements. Similar data growths are being witnessed in other storage databases besides those for DNA data. For example, one of the goals of the iDigBio project is to digitize close to a billion specimens housed in various museums and herbaria in the United States. This is more than the current specimen records on the site [29].

We cannot discuss the changes of biodiversity big data curation without acknowledging the role played by cyber infrastructure (CI) development. In the past decade or so there has been significant development and improvement in technological advances especially in computer infrastructure. The development of powerful tools in line with improving cyber infrastructure (CI) has helped to create the space for the storage of high volume of data with minimal problems. The changes brought about by current developments and improvements in cyber infrastructure have seen some of the big data platforms storing and curating data in such a way that data can be linked to the relevant analytical tools. This then ensures a quick and efficient use of data, which is important for evolutionary biologists, taxonomists, ecologists and other biodiversity data users. For example, The Open Tree of Life database is continuously improving and evolving with shifts in the types of questions asked by researchers making it an important tool for evolutionary biology [29]. There is no doubt that technological advances in CI help to facilitate new and innovative research using the stored data, ensuring a successful future in biodiversity research and conservation. The biodiversity community will continue to benefit from the advances in CI development.

### 3.1. Opportunities in Data Storage and Curation

Linkages between some of the big data sites such as Open Tree of Life and iDigBio together with relevant cyber infrastructure and several other tools, for example the BiotaPhy project, allows researchers to address different evolutionary questions very quickly [29]. The different types of data allow integrative research on biodiversity which in turn gives a starting point for evaluating the effects of environmental problems such as invasive species and the impacts of climate change. With the right infrastructure and improved analytical methods, it is possible to combine genetic, morphological, and other trait data from big datasets to undertake a comprehensive set of analyses. For example, Map of Life, is an e-infrastructure tool that uses data from GBIF records to spatially connect point data with layers of conservation reserves and geographical ranges. Therefore, when integrated with good e-infrastructure, big data can be analyzed to facilitate quick and informed decision making [28].

### 3.2. Challenges in Data Storage and Curation

Although progress has been made in the development of analytical tools and cyber infrastructure for handling big data, there is still room for further improvement. Some big data are not readily available. In other cases, the data entry and retrieval formats are difficult to understand, thus putting off potential users.

Often, there are some inconsistencies in how data is curated, especially when it involves citizen scientists who may lack the technical skills to correctly identify the biota, for instance, or are unable to distinguish between nomenclatural synonyms. All these lead to data inaccuracy. Storage of data can be very expensive especially when it comes to the maintenance of stored data and the upgrading of the systems where the data is stored. In cases where physical collections are converted to digital formats, they lose part of their ecological meaning since some relevant ecological information is exclusively found in the physical records rather than on databases [10].

The constant evolution of data storage platforms comes with the challenges of having to develop or keep modifying dependent analytical tools. In some cases, due to the differences in data types and standards, it can be challenging to integrate different datasets into a single analysis workflow [28]. At the same time, the community that utilizes such platforms must keep up with these changes and upskill for the technical competencies required to navigate the system.

### 3.3. Recommendations in Data Storage and Curation

Although big data platforms such as the Global Biodiversity Information Facility have gone a long way in serving the scientific community on different levels, there is still room for improvement to maintain the reliability, credibility and accuracy of data found on such platforms. For example, many of the GBIF's occurrence records of over 1.7 billion specimens are not represented by voucher specimens [29], thus indicating a need to develop validation tools for this platform. Whilst we recognize the value of a georeferenced information for a specimen in making conservation decisions, having a standardized framework for data storage and curation will improve the accuracy and reliability of data stored on the different databases. Putting in place systems that check for consistency between new data and already existing data on storage platforms as well as detecting any outliers to minimize errors are other ways that can be deployed to improve existing systems.

## 4. Access to Biodiversity Data

The accessibility of biodiversity data from the different big databases can be classified at different levels, namely: unrestricted; restricted for confidentiality purposes; require permission to access; or require formal acknowledgement first [49]. It is vital to promote free and open use of this data for many good reasons. However, ready access to data is not always easy or possible [32], and historically there has been a generic culture of not

sharing science data [50]. If a database includes information on locations of threatened or rare species, then accessibility to such data may be restricted for the purpose of protecting such species [49]. Some restrictions are put in place to generate money from those who will be granted access after payment. In some cases, custodians of the data restrict accessibility to protect ownership of the data especially for research purposes, while some are reluctant to share due to lack of incentives, rewards or other forms of recognition [32].

A review of the Australian ERIN database found that 51% of data on the database were confidential data, restricted data or data requiring permission for accessibility, while 49% were freely accessible [51]. Access to biological data is more restricted relative to other biodiversity-related data types, for example, environmental data [32]. Although efforts are underway to digitize collections in natural history museums and herbaria, a significant amount of taxonomic data on plants still exist as physical specimens and in paper copies rather than in a readily accessible digital format [52]. Until recently, accessibility to data in natural history museums has mostly been limited to curators, taxonomists, and researchers in biosystematics. With the expectation of the continuous increase of digitization of natural history collections, constraints to accessibility of specimen data are expected to decrease [53].

The increase in threatened habitats, which may get worse in the coming decades if current projections are anything to go by, and the challenges of climate change are some the reasons for data accessibility to be more open. Research undertaken with such data can inform better decision for the protection of biodiversity. Restricting access to biodiversity data can end up being one of the limitations to achieving global conservation goals.

On a brighter note, it is encouraging that calls have been made and several concepts are being developed to promote the freeing up of data and encourage data sharing. This has resulted in open access data sharing concept being widely adopted and declared as best professional practice [47]. This is made easier in the current era, where digitization is being embraced, and global access to the internet is becoming the norm [53]. The Rio Convention of 1992 has been instrumental in the progress towards the free and open access of science data [52], resulting in big data platforms such as GBIF adopting and implementing this approach [47]. The expected trend in coming years is that more databases will go the open-source route.

Two major projects, summarized below, highlight the value of data sharing (through collaboration) and open access biodiversity data. The PREDICTS project [54] was built by collating freely shared data from a large collection of quality assured empirical studies across biodiversity science and integrating the massive dataset with remotely sensed climatic data. One of the main goals of the project is to provide a better understanding of the impact of biodiversity loss on ecosystem functions and services. The project, which is dynamic as it continues to incorporate more relevant data, has developed its own database now being used to generate high quality models for understanding human impacts in relation to various land use practices across the globe. The other project [55] investigated global tree species distributions by combining data from five aggregators of the occurrence data, including GBIF. The project distilled the big dataset into categories of data quality and used high quality records to generate robust model of tree species distribution. The work also shows geographical areas of data gap and the need for data quality improvement processes. Without ready access to BD, the idea of implementing projects of this magnitude would not have taken off in the first place.

### 4.1. Opportunities in Access to Biodiversity Data

Data access can help promote and accelerate the development of innovative solutions in biodiversity management. Open data can increase knowledge creation using existing knowledge base through research. It can also increase and encourage collaboration among several stakeholders at different levels, from those who collect the data, those who utilize it for research and to those who use the results of the research to make decisions and formulate policies that are biodiversity related. This promotes the sharing of data, reusing

data and improving data quality by users who now have vested interest in good data they can always access. On a large scale, multidisciplinary and interdisciplinary research collaborations can be built for mutually leveraging each other for greater efficiency and accelerated development to benefit all aspects of the biodiversity enterprise.

### 4.2. Challenges in Access to Biodiversity Data

Some datasets are behind paywall, rendering such data inaccessible to organizations and individuals without the financial resources. Even for those who could pay, ethical and perhaps idealistic considerations (e.g., why should anyone pay to access data generated through publicly funded research?) may prevent them from accessing the data. Another challenge to data access is the difficulty in locating some data repositories. This is easily underestimated by developers of data platforms and those who deposit data there. If websites and data platforms are not published and the links widely circulated to the public, locating the repositories becomes a major hurdle to data access even if the datasets are freely available. In some cases where a data platform is well-known, there are no clear and easy-to-follow guidelines on how to access the datasets of interest. With respect to data housed in physical museums or herbaria, access is automatically limited to those who can be in the physical space, thus locking out those who are unable to afford the logistical cost of visiting the repository.

Another barrier to data access is tied to the behavior of some data creators in hoarding their data or prevent access for a specified period. While the accessibility to data is desirable, these behaviors are understandable from the perspectives of the data creators to promote proper attribution on the one hand, and to avoid being scooped on important insights from their data on the other. By and large, inaccessible data constitutes a body of information that is not widely available for many interested parties to use.

### 4.3. Recommendations in Access to Biodiversity Data

Although it is now being widely discussed in scientific circles, there is need for more ways to acknowledge and incentivize data creators. Protocols can be developed and widely promoted on the issues of acknowledging the owners of the data we utilize. The current practice of floating scientific journals dedicated to the publication of raw data and containing links to where they are stored is a good starting point to encourage data sharing. It is equally helpful to accelerate the mobilization of data into online repositories. Researchers and organizations involved in generating data should be encouraged to have parallel digital curation for all data collected and stored as hardcopy. This would minimize the struggle with accessing data stored either on disparate local computers, or in museums thousands of kilometers away from the end user. Data that is available in digital form is much easier to access as such data are just a click away.

### 5. Data Analysis

The goal of data collection endeavors is to derive value from datasets to guide decision making and necessary action plan. Deriving such value is at the heart of data analytics, which has become a big industry on its own. Any set of data, however big, is meaningless until and unless insight is extracted from it through an appropriate set of analyses. The need to manipulate large datasets in biodiversity science across various platforms has spawned the relatively young discipline of biodiversity informatics [56]. Here, we adopt the broad scope that analysis should encompass a set of "[w]ell-governed interoperable e-infrastructure, and workflows should support biodiversity discovery and documentation, environmental monitoring, reporting and decision making, as well as the capacity to run fundamental scientific modelling experiments to build understanding of biodiversity evolution, biogeography, and dynamics in a changing world" [28]. This view resonates with the four ideals of data analytics, which are description, explanation, prediction and prescription [57]. However, as a starting point for biodiversity data analysis, datasets need to be prepared to render them into useable and compatible formats for the required set

of analyses. These pre-analytics steps involve data selection from a variety of sources, pre-processing to remove/reduce noise, dimension reduction through transformations, and finally enrichment by combining with other complementary datasets to provide better insight into the questions at hand [5,58].

Given the disaggregated and complex nature of many biodiversity datasets, the disparities in scale of observations and the variation in sampling techniques and differing research purposes, a major challenge facing BBD analysis is the mobilization and integration of these datasets into a coherent whole that is fit-for-purpose [29,59]. Relevant to the integration step is the development of cyber infrastructures, biodiversity analytical platforms and synergistic automated workflows to afford researchers the time to focus on doing their science. Recent developments in biodiversity informatics are largely encouraging as the acquisition of such integration facilities are gradually being prioritized, and the potential they hold for solving real-life biodiversity problems is being demonstrated by various case studies [26,54,55].

Some of the tools available for undertaking robust large scale biodiversity analysis include Lifemapper, which uses species occurrence records (available online) to produce distribution maps and makes prediction of habitat suitability for any given species based on the occurrence records [60]; BiotaPhy works on similar principles as Lifemapper [61]; Infomap bioregions and SpeciesGeoCoder [62] use species distributions data to assess both current and historical spatial groupings of taxa that could be important for conservation decision making [63], and ancestral area reconstruction; SUPERSMART [64] is a platform for assembling molecular and fossil data, and inferring robust time-calibrated phylogeny for any group of taxa. All these tools have the potential for hypothesis-driven research in historical biogeography, conservation, and systematics. Soltis et al. (2016) [29] offered a summary of other big data analytical tools in biodiversity, and detailed potential workflows for cross-linking them to address several big questions in biodiversity science. A recent and still ongoing advancement in biodiversity is the development of an analytical framework to interface primary biodiversity observations, indicators and assessment possibilities [65]. The Essential Biodiversity Variables (EBVs) framework [66], as it is referred to as, is a coordinated means to quantify biodiversity dynamics on a global scale, reducing the complexity of biodiversity into a list of priority measurements [67]. The framework is theory-driven rather than data-driven, helping to strengthen the information basis of biodiversity reporting to guide policy instruments [65]. The concept of EBVs is already finding application in monitoring both the populations of single species or their aggregates at multiple spatial scales of relevance to diverse research questions and associated decision-making [3].

A major debate at the core of big data analysis, especially of biodiversity, is the pre-eminence ascribed to the pattern-recognition powers of algorithms, usually to the abandonment of hypothesis testing and theory formulation. However, "[f]raming the issue of Big Data in terms of oppositions, that is, deduction versus induction, hypothesis-driven versus data-driven or human versus machine, misses the point that both strategies are necessary and can complement each other" [68]. Pattern detection capabilities driven by machine learning, artificial intelligence and related algorithms, can be the basis of fine-tuning research questions, hypothesis testing and new theory development. While it is important to put powerful analytical tools within the reach of researchers, meaningful data analysis still requires a clear circumscription of problems to which the analytical method is tailored, and for which the dataset is well suited.

The development of big data analysis in biodiversity, or any field for that matter, has a very young history, tied to the development of powerful computers and algorithms to match. More recent developments in the use of biodiversity big data, which have resulted in the emergence of new data sources and cyber infrastructure for organizing and integrating large biological datasets, have prompted the improvement of big data analytics [29]. Equally, historically, before the development of e-infrastructure data aggregators and servers, scientists have been striving to improve the techniques needed to analyze

big data [45]. Although there are limitations in the currently available infrastructures for biodiversity big data analysis, the prospects of developing appropriate solutions are encouraging [69].

### 5.1. Challenges and Opportunities in BBD Analytics

The challenges of BBD analytics summarized here focus on the scientific side of big data rather than the financial resources for procuring the required infrastructures. Regardless of the power of analytics and the size of the data, the quality of insight derivable from any analysis is a function of the fitness of the dataset(s) for the questions of interest, all other things being equal. There are clear limits to what analytics can discern from poor quality datasets or the wrong use of datasets. In addition, it has been shown that an abundance of data for a particular purpose does not necessarily translate to more knowledge, as the data may be unstructured, such as those collated from citizen science initiatives and remote sensing technologies [70]. Despite the massive number of biodiversity datasets at our disposal, now more than at any other time in history, there are some inevitable shortfalls in our knowledge base, resulting in trade-offs between generalities and uncertainty, thus constraining the value derivable from available big data [55,71]. The reality about data gaps is that while we can reduce them to answer certain questions [72], we can never truly fill them all due to logistical and financial constraints. Closely linked to the wrong application of analytical method is the issue of technical expertise to analyze BBD meaningfully. The development of many open access biodiversity analytical platforms is making automated analysis relatively easy [29]. Nevertheless, there are significantly more opportunities to analyze large datasets than the volume of analyses being undertaken suggesting, among other possibilities, that a limited number of people possesses the skill set required to utilize currently available workflows [73]. It is equally plausible that the human factor of reluctance to embrace change is at play [59], given the disruptive nature of big data and its associated analytical tools. Finally, with an increasing number of data aggregator facilities, and the dynamic nature of BBD that keeps getting bigger, compatibility among platforms is a potential problem that could slow down development.

These challenges notwithstanding, the analysis of BBD presents opportunities to foster collaborative engagements across the various domains of biodiversity, used to operating within their disciplinary silos. It also opens avenues for technological innovations. Most of the key infrastructural components, both in terms of hardware and software, are already available [28]. The complexities of biodiversity science and the need for solutions beyond the capabilities of any singular organization or discipline is, rightly, leading to coordinated efforts on a global scale in providing a systems-level response to the biodiversity crisis.

### 5.2. Recommendations in BBD Analytics

To ensure quality analyses that can help with effective decision making and policy formation, data quality control processes must be in place. Human capacity development to use existing technologies [59], and regular upskilling due to a rapidly changing analytical landscape is vital. There is also an urgent need for the development of purpose-specific rankings of datasets and improved analytical models that account for data gaps [70].

## 6. Communicating Biodiversity Science to Inform Policy Formulation

The currently accepted view of science communication is that of an ongoing dialogue where science interacts with the public and other stakeholders in a multi-way stream of engagement [74]. This contrasts with the deficit model of science communication which supposes that provision of facts is sufficient for decision making and behavioral change [75]. Indeed, the consensus is that people's interpretation of science is influenced by their culture, ethics and other filters independent of the scientific fact at hand [76,77]. Given this background, it is little surprising that our increased knowledge of biodiversity is not on par with biodiversity policy guidelines and decisions.

Globally, biodiversity loss continues unabated, especially in ecologically valuable areas [78]. This is despite our wealth of knowledge accumulated from massive datasets, thus supporting the perception that insight from biodiversity science is underutilized in policy formulation [79]. A review of the literature at the interface of science communication and policy identified the linear model of science–policy interaction as a major impediment in translating good science into sound policy [80]. The model assumes that science and policy belong in separate domains and are treated as such, with science purportedly providing accurate answers to well-defined questions of policy makers. Available evidence, however, suggests that policy formulation is a much more nuanced process and is influenced by several considerations, of which scientific merit is but a fraction [80].

A cursory survey of how biodiversity research findings are being communicated indicates that most outputs are published in peer-reviewed science journals or technical books and volumes, automatically restricting the audience to fellow scientists. Added to this, the majority of science journals are locked behind pay walls, thus further limiting access to biodiversity research even among practitioners [81]. Several other discoveries are presented in learned conferences, which are largely gatherings of experts in the field. These communication practices amount to preaching to the biodiversity choir. Many policy makers are non-scientists whose understanding of biodiversity is shaped by readily accessible pieces (with their sensational and misconstrued headlines) in the popular media and not from scientific journals. For knowledge to shift mind-sets, therefore, the onus is on scientists to device effective means of conveying their hard-won findings to policy makers. Recent developments around the communication of biodiversity research recognize this need, and several calls-to-action have been issued to give effective communication a prominent role [82,83]. Tested strategies that have been proposed and are being deployed to bridge the communication divide include deliberately targeting categories of stakeholders outside the ivory towers with relevant information [82,84]. The media is of particular interest here because of the critical role they play in framing issues, and their power for influencing the direction of public policy. Legagneux et al. (2018) [85] highlighted the role of non-scientists in drawing global attention to climate change crisis through massive media coverage and involvement of global public figures to champion the cause. However, whether enough is being done by the protagonists of biodiversity science to close the gap between the science and its communication to influence policy remains an open question. In summary, for biodiversity science communication to achieve its goals, it might need to borrow from advances in communication and apply it as a developmental tool. It must view communication as an ongoing process of reciprocal interchange of views and opinions between the science and the public [74,86].

## 6.1. Challenges and Opportunities in Biodiversity Science Communication

Biodiversity science is widely recognized as complex and its communication to lay audience is no less. This, combined with the fact that many scientists are not trained in science communication and have, therefore, never thoughtfully entertained the prospect of breaking down their research to the non-scientist. Another factor that can muddle the communication waters between biodiversity science and stakeholders is a lack of understanding, on the part of the public, of the bounded uncertainties inherent in many biodiversity research, leading to unrealistic expectations of what science can deliver. Furthermore, biodiversity scientists (as are all scientists), are not always neutral parties on a particular policy issue. They sometimes hold biased views on which side they advocate for policy-wise; at other times, they operate under considerable political pressures. One other barrier to effectively communicating about biodiversity is the problem of assessing, in quantitative terms, the value of biodiversity. This is because not all values derivable from biodiversity (e.g., aesthetics) can be readily translated into quantitative formats [79,87].

The challenges present opportunities to develop, test and implement strategies for effectively conveying the key messages of science to stakeholders. For instance, development of inter-disciplinary studies at the interface of science and policy might create a unique

category of professionals straddling both worlds comfortably to drive necessary policy transformations and biodiversity agendas.

*6.2. Recommendations in Biodiversity Science Communication*

Scientists should adapt scientific communication methods to other people's world view and form partnerships with non-scientists including the media to minimize miscommunications. Involvement of well-known global figures as biodiversity champions will go a long way to get the public and could potentially promote positive media coverage. There should also be a management of expectations as to the extent of the contribution that science can really make to wise biodiversity decision-making process [88].

## 7. Synthesis and Conclusions

The continuous loss of biodiversity affects ecosystem functioning, of which we are a part. To stem the tide, evidence-based decision-making processes should become the normative mode of operation. This is only possible on the back of adequate and quality data that is well analyzed and accurately interpreted. This review presents the data life cycle as an umbrella framework for critically engaging the subject of big data in biodiversity science with the goal of making informed decisions in biodiversity management. Although we present the framework in what appears to be a logical flow starting from data generation, through storage, to analysis and finally to communication, any of the themes could, arguably, be a starting point for engagement depending on context. The themes and associated sub-themes are all interlinked and dependent on each, and not necessarily in the neat order we have arranged them. Data collection could be informed by the analysis of previously available datasets, which may identify specific data gaps. In turn, data analysis is underpinned by access to some sets of data in the first place. For informed policy decisions on biodiversity issues, the insight gained through analysis must be effectively communicated to stakeholders and policy makers. Infrastructural developments to drive innovative data collection, the storage of massive datasets and the performance of relevant analyses are critical to the smooth operation of the scheme. The interlinked nature of the scheme suggests that there will be some element of redundancies for quality assurance. As summarized in Figure 2, such overlaps are reflected in the similarity of challenges and opportunities across some themes.

| | Generation & Collection | Storage, Access & Curation | Analysis | Communication |
|---|---|---|---|---|
| **Challenges** | Errors and inaccuracies in data (poor quality data). Biased data collection resulting in data gaps for some taxa or regions. Time consuming, expensive, and labour intensive. Incompatible data formats and platforms | Lack of access to some datasets. Prohibitive cost of accessing the physical spaces housing biodiversity data and associated specimens. Financial cost of storage; some data behind paywall. Inconsistencies of style and format. | Inappropriate analysis due to the ease of point-and-click tools. Scarcity of experts in big data analysis with a sound grasp of biodiversity. Incompatibilities among platforms. | Miscommunication (by the scientist) and misunderstanding (sometimes by the public). Quantitative assessment of the value of biodiversity. Biased view of the proponents of biodiversity science. Political pressure. |
| **Opportunities** | Interdisciplinary collaborations. Development of infrastructures for data aggregation. Automated collection of high-volume data in some domain. | Innovative solution through integration of different datasets. One-point stop to access, integrate and analyse several datasets. Reusable data. Research leveraging through multidisciplinary collaboration. | Development of intelligent algorithms for big data analysis. Rapid decision based on robust data analysis. Development of system-level solution. Multidisciplinary collaboration. | Test of effective communication strategies. Development of professionals to straddle the science-policy interface. |
| **Recommendations** | Adequate training for data collectors. Development of data collection protocols to promote consistency and improve accuracy. | Development of data validation tools. Standardized format of storage and curation of data of similar type. Creative ways to incentivise data owners. | Training of biodiversity researchers to use available tools. Development of improved analytical models. | Provision of regular science communication training to scientists. Recruitment of well-known figures as biodiversity advocates. |

**Figure 2.** Summary of challenges and opportunities across BBD themes.

The cyclic nature of the scheme also connotes the potential reusability of biodiversity data. Indeed, this is a necessity due to the historical element inherent to biodiversity

datasets, and the logistical and financial constraints of data collection. Because biodiversity scientists are usually directly involved in every theme of the scheme except, perhaps, for the policy formulation and decision-making phase, the need for deliberate constructive engagement between scientists and policy makers becomes non-negotiable. A good starting point for such engagement is the recognition by both sets of players that they belong in the same domain, even if their roles are different. Critical to those roles is good quality big data and what can be done with it.

**Author Contributions:** Conceptualization, A.A. and T.M.; Theme definition, A.A., T.M. and M.G.D.; Literature search and compilation, T.M. and M.G.D.; Writing—original draft preparation, T.M. and A.A.; Writing—review and editing, A.A., T.M., M.G.D.; Visualization, A.A.; Project administration, A.A.; Funding acquisition, T.M. All authors have read and agreed to the published version of the manuscript.

**Funding:** This research was funded by ABSA Bank, grant number 11112019 and The APC was funded by ABSA Bank.

**Acknowledgments:** The authors thank Guy Midgley for reading through and offering helpful suggestions on an earlier draft of the manuscript.

**Conflicts of Interest:** The authors declare no conflict of interest. The funders had no role in the design of the study; in the collection, analyses, or interpretation of data; in the writing of the manuscript, or in the decision to publish the results.

## Appendix A

**Table A1.** Examples of notable biodiversity big data platforms.

| Platform/Site | Type of Data | Number of Records | Reference/Website |
|---|---|---|---|
| GenBank | Nucleotide sequences and their protein translations | >2 billion sequence records | www.ncbi.nlm.nih.gov/genbank/, accessed on 10 May 2021 |
| Barcode of Life Data System (BOLD) | DNA barcode sequences | >6 million DNA barcode sequences from over 542,000 species. | http://barcodinglife.org/, accessed on 8 April 2021 |
| Global Biodiversity Information Facility (GBIF) | Specimen-based and observational data on localities | >1.6 billion records | gbif.org, accessed on 8 April 2021 |
| Integrated Digitized Biocollections (iDigBio) | Digitized neontological and paleontological biodiversity collections and associated media and metadata, specimen location | >70 million specimen records | www.idigbio.org, accessed on 8 April 2021 |
| The Atlas of Living Australia | Collaborative, digital and open infrastructure that pulls together Australian biodiversity data from multiple sources, making it accessible and reusable | >67 million records | https://www.ala.org.au/about-ala/, accessed on 7 April 2021 |
| The Open Tree of Life | Phylogenetic data and genealogical tree connection for all of Earth's >2.3 million named species | >2.3 million of earth's named species | blog.opentreeoflife.org, accessed on 10 April 2021 |
| Chinese Virtual Herbarium | Records from the flora of China | >3 million records | http://www.cvh.org.cn/, accessed on 10 April 2021 |
| Digitized herbarium of the Museum National d'Histoire Naturelle (MNHN) in Paris | Collection of vascular plants | >5 million records | https://science.mnhn.fr/institution/mnhn/collection/p/item/search, accessed on 10 April 2021 |
| Australia's Virtual Herbarium | Specimen records of plants, algae and fungi | >7 million records | http://avh.chah.org.au/, accessed on 10 April 2021 |

**Table A1.** *Cont.*

| Platform/Site | Type of Data | Number of Records | Reference/Website |
|---|---|---|---|
| Institutos Nacionals de Ciencia e Tecnologia e Herbario Virtual da Flora e Dos Fungos | Digitized specimen records | >5 million records | http://inct.florabrasil.net/, accessed on 14 March 2021 |
| Canadensys | Digitized specimen and occurrence records especially for plants, insects, and fungi | About 3 million records | https://community.canadensys.net/, accessed on 20 November 2020 |
| JACQ Virtual Herbarium | Digitized specimen records | >5.5 million specimens | http://herbarium.univie.ac.at/database/index.php, accessed on 20 November 2020 |
| LUOMUS | Digitized botanical and mycological collections | >9 million specimens and sample lots | www.luomus.fi/en/botanical-andmycological-collections, accessed on 23 November 2020 |
| Encyclopedia of Life Trait Bank | Trait data records for different taxa | >11 million records for over 330 attributes for more than 1.7 million taxa | www.eol.org/traitbank, accessed on 30 October 2020 |
| TRY Plant Trait Database | Trait record data for plant species | >5.6 million trait records from more than 100,000 plant species | www.try-db.org, accessed on 20 November 2020 |
| GloBI (Global Biotic Interactions) | Species interaction data | >1.3 million interactions for over 113,000 distinct taxa | www.globalbioticinteractions.org/about.html, accessed on 20 November 2020 |
| Catalogue of Life | World's most comprehensive and authoritative index of known species of animals, plants, fungi and micro-organisms | 1,829,672 living and 38,145 extinct species | www.catalogueoflife.org, accessed on 23 November 2020 |
| International Barcode of Life (iBOL), | Use of sequence diversity, standardized gene regions as a tool for identifying known species and discovering new ones | >5 million georeferenced records | www.ibol.org, accessed on 10 April 2021 |
| Australian Environmental Resources Information Network (ERIN), | Environmental information and data | | https://www.environment.gov.au/about-us/environmental-information-data/erin, accessed on 10 April 2021 |
| UK Biological Records Centre (BRC), | Focus on UK terrestrial and freshwater species records | Unknown | https://www.brc.ac.uk/, accessed on 10 April 2021 |
| US Gap Analysis Project | Species, land cover and protected areas database of the United States | Unknown | https://www.usgs.gov/core-science-systems/science-analytics-and-synthesis/gap, accessed on 10 April 2021 |
| Index Herbariorum | Herbaria serving species of bryophytes, ferns, lycopods, gymnosperms, and angiosperms | 3400 herbaria with 350,000 species and 350 million specimens | sweetgum.nybg.org/science/ih/, accessed on 11 April 2021 |
| National Biodiversity Network Gateway | Collects, sorts, analyses, and disseminates data for biodiversity in the United Kingdom | >127 million species records | http://data.nbn.org.uk/, accessed on 11 April 2021 |
| Biodiversity Data Centre | Data and information on species, habitat types and sites of interest in Europe | Unknown | http://www.eea.europa.eu/themes/biodiversity/dc, accessed on 10 April 2021 |

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
