# Peer review of "Big Data in Biodiversity Science: A Framework for Engagement"

_technologies, doi:10.3390/technologies9030060_

Round 1

Reviewer 1 Report

This manuscript try to give a general review for the Big Data in Biodiversity, but just show universal data management issues in biodiversity, not showing the real opportunity and challenge for "Big Data". The content has no creative suggentions for current big data management in biodiversity or in local scale study. Try to make clear comparation between the big data issue and the tranditional data management issues in the future. And maybe more tables and figues can help to organize the paper clearly. 

Author Response

We have provided an additional figure/table to organise the paper clearly, as suggested. Our figure 2 made clear recommendations regarding each of the themes we identified. These recommendations cover the data management as well. Because the scope of big data literature is wide, we defined our objectives towards the concluding part of the introduction, and we worked within those objectives.

Reviewer 2 Report

Referee’s Report on

“Big Data in Biodiversity Science: A Framework for Engagement”

by

Tendai Musvuugwa, Muxe Dlomu and Adekunle Adebowale

Summary

In this review paper, the authors present the life cycle of data as a framework to interrogate four main objectives: i) summarise the current state of Biodiversity Big Data under each theme of the scheme, ii) identify opportunities for innovation/development/collaboration, iii) identify challenges, and iv) propose recommendations to drive best practices in the business of Biodiversity Big Data.

General comments

The paper deals with an interesting topic (big data is a hot top nowadays), as is the engagement of big data in biodiversity science. The paper is well-written and covers all aspects. can be further improved.

Improvements that you could suggest on the paper

In Figure 1, the authors present a diagram of the life cycle of biodiversity data as applied in the paper. This figure is clearly helpful at the beginning of the paper. However, in my opinion, another similar diagram, which will summarize what was discussed in the paper would help the reader to retain the main points/conclusions of this review paper.

Author Response

As suggested by this reviewer, we have provided a second figure to summarise the key points of the paper. We are grateful for the helpful input.

Reviewer 3 Report

Comments for the manuscript:

Big Data in Biodiversity Science: A Framework for Engagement

This manuscript is a review of data science, communication, and decision-making in the context of biodiversity science. The paper also proposes a framework to support those activities and take evidence-based decisions, including recommendations and best practice guidelines.

The subject of the manuscript is current and very important.

The framework proposed can be helpful by providing to different actors (with diverse scientific backgrounds) working on this problem a uniform basis for using these tools in decision-making and communication.

The paper is well written and easy to read. The review is extensive and relevant. In my opinion, the manuscript attains its objective and could be very relevant if the proposed methodology and framework take root in the biodiversity science community.

The paper needs some English changes. For example, I think there are missing or wrong articles before the words: "efficient" in line 11, "vast" in line 12, "purpose" and "informed" in line 75, etc. 

Author Response

We have corrected all the missing and wrong articles pointed out by the reviewer. We have effected other changes to make the paper easier to read and understand. Thank you for your helpful suggestions.